# Accuracy of CT perfusion ischemic core volume and location estimation: A comparison between four ischemic core estimation approaches using syngo.via

Jan W. Hoving[1]*, Miou S. Koopman[1], Manon L. Tolhuisen[1,2], Henk van Voorst[1,2], Marcus Brehm[3], Olvert A. Berkhemer[1], Jonathan M. Coutinho[4], Ludo F. M. Beenen[1], Henk A. Marquering[1,2], Bart J. Emmer[1], Charles B. L. M. Majoie[1]

1 Department of Radiology and Nuclear Medicine, Amsterdam UMC, Location University of Amsterdam, Amsterdam, The Netherlands, 2 Department of Biomedical Engineering and Physics, Amsterdam UMC, Location University of Amsterdam, Amsterdam, The Netherlands, 3 Siemens Healthcare GmbH, Computed Tomography, Forchheim, Germany, 4 Department of Neurology, Amsterdam UMC, Location University of Amsterdam, Amsterdam, The Netherlands

* j.w.hoving@amsterdamumc.nl

**Data Availability Statement:** Individual patient data cannot be made available under Dutch law as the authors did not obtain patient approval for

## Abstract

### Background and objective

Computed tomography perfusion (CTP) is widely used in the evaluation of acute ischemic stroke patients for endovascular thrombectomy (EVT). The stability of CTP core estimation is suboptimal and varies between software packages. We aimed to quantify the volumetric and spatial agreement between the CTP ischemic core and follow-up infarct for four ischemic core estimation approaches using syngo.via.

### Methods

We included successfully reperfused, EVT-treated patients with baseline CTP and 24h follow-up diffusion weighted magnetic resonance imaging (DWI) (November 2017–September 2020). Data were processed with syngo.via VB40 using four core estimation approaches based on: cerebral blood volume (CBV)<1.2mL/100mL with and without smoothing filter, relative cerebral blood flow (rCBF)<30%, and rCBF<20%. The follow-up infarct was segmented on DWI.

### Results

In 59 patients, median estimated CTP core volumes for four core estimation approaches ranged from 12–39 mL. Median 24h follow-up DWI infarct volume was 11 mL. The intraclass correlation coefficient (ICC) showed moderate–good volumetric agreement for all approaches (range 0.61–0.76). Median Dice was low for all approaches (range 0.16–0.21). CTP core overestimation >10mL occurred least frequent (14/59 [24%] patients) using the CBV-based core estimation approach with smoothing filter.

sharing individual, coded patient data. In line with privacy regulations, publication of individual patient data as well as syntax files and output of statistical analyses is forbidden by the Data Privacy Officer of the Amsterdam UMC. All syntax files and output of statistical analyses are available on reasonable request. Such requests can be addressed to Matthan W. A. Caan at m.w.a.caan@amsterdamumc.nl.

**Funding:** The author(s) received no specific funding for this work.

**Competing interests:** We have read the journal's policy and the authors of this manuscript have the following competing interests: HAM is co-founder and shareholder of Nicolab. BJE reports grants from Stryker Neurovascular and personal fees from Dekra and Novartis outside the submitted work. CBLMM is shareholder of Nicolab and reports grants from TWIN, CVON/Dutch Heart Foundation, and Stryker outside the submitted work. This does not alter our adherence to PLOS ONE policies on sharing data and materials. All other authors declare no support from any organization or financial relationships with any organizations that might have an interest in the submitted work. All authors declare no other relationships or activities that could appear to have influenced the submitted work.

## Conclusions

In successfully reperfused patients who underwent EVT, syngo.via CTP ischemic core estimation showed moderate volumetric and spatial agreement with the follow-up infarct on DWI. In patients with complete reperfusion after EVT, the volumetric agreement was excellent. A CTP core estimation approach based on CBV<1.2 mL/100mL with smoothing filter least often overestimated the follow-up infarct volume and is therefore preferred for clinical decision making using syngo.via.

## Introduction

Computed tomography perfusion (CTP) is widely used in the evaluation of acute ischemic stroke patients for endovascular thrombectomy (EVT) [1–3]. CTP is also used to identify patients who are eligible for IV alteplase treatment when EVT is contraindicated or not planned between 4.5–9 hours after stroke onset [4, 5]. CTP could quantify the cerebral perfusion based on the following parameters: cerebral blood volume (CBV), cerebral blood flow (CBF), mean transit time (MTT), time-to-maximum (Tmax), or time-to-peak (TTP). With these parameters, the extent of the severely hypoperfused 'ischemic core' and the hypoperfused, but–if timely reperfused–viable 'penumbra' can be estimated. Although the use of CTP ischemic core volume for the selection for EVT of patients in the 6-24h time window is included in the current guidelines, it is not recommended for selection in the 0-6h time window [6, 7]. Yet, CTP is still commonly performed in the 0-6h time window in the Netherlands, albeit usually without clinical consequenses. The various commercially available CTP post-processing software packages use different approaches based on CBV or relative CBF (rCBF), and Tmax or rCBF parameters thresholds to estimate the respective ischemic core and penumbra, which complicates the generalizability of CTP results [8–11]. Siemens syngo.via (Siemens Healthcare, Erlangen, Germany) is a widely used CTP post-processing software package, with a recommended CBV-based core (threshold CBV <1.2 mL/100mL) and CBF-based penumbra (threshold CBF<27 mL/100mL/min) estimation approach. Another frequently used CTP software package is RAPID (iSchemaView, Menlo Park, CA, USA), which uses relative CBF (rCBF) <30% and Tmax >6 seconds as the thresholds for ischemic core and penumbra, respectively. Previous studies have focused on differences between different CTP post-processing software packages and the comparison of the CTP results between vendors based on modifying specific thresholds [8, 10–19]. However, the agreement of core estimation for different CTP ischemic core estimation approaches using syngo.via has sparsely been studied [8, 12, 13]. Moreover, it is unclear how different core estimation approaches affect the accuracy of CTP in estimating the 24h diffusion-weighted imaging (DWI) follow-up infarct lesion. This is clinically important, since the follow-up infarct volume is, inter alia, a strong predictor of functional outcome [14]. We aimed to determine the extent of differences among four commonly used CTP core estimation approaches by quantifying the volumetric and spatial agreement between the CTP ischemic core and the 24h follow-up DWI infarct using syngo.via.

## Materials and methods

### Study population

We performed a single center, single acquisition protocol retrospective analysis of a prospectively collected registry of EVT-treated patients with baseline CTP. We included patients who

were presented to our center for EVT between November 2017–September 2020. Other inclusion criteria were: admission within 24 hours after symptom onset, present occlusion of the anterior circulation, successful reperfusion (defined as expanded treatment in cerebral ischemia (eTICI) score 2b-3), and available 24h follow-up diffusion weighted imaging (DWI). Patients were not included for analysis if the CTP data could not be made available due to acquisition and storage at the primary stroke center according to the European General Data Protection Regulation (GDPR).

## Image acquisition

CTP images were acquired on a dual source 192-slice scanner (70 kVp, 12 cm coverage; SOMATOM Force, Siemens Healthcare, Forchheim, Germany) with a slice thickness of 1mm and 0.7mm increment. All acquisitions were reconstructed to 5mm slices. All CTP scans were acquired after intravenous injection of 35 mL iodinated non-ionic contrast agent (Iomeron 300, iomeprol, 300mg iodine/mL; Bracco Imaging Deutschland GmbH, Konstanz, Germany) with an injection rate of 6 mL/s. CTP data were acquired using the following protocol: 15 scans 1.5 seconds apart, followed by 15 scans 3 seconds apart, resulting in a total of 30 scans over a period of 60 seconds. Baseline non-contrast CT and CT angiography images were acquired on the same dual source CT scanner, with the CTP acquired after the non-contrast and before the CTA. Depending on scanner availability, follow-up MRI DWI (b = 0 s/mm$^2$ and b = 1000 s/mm$^2$) and apparent diffusion coefficient (ADC) images were acquired on a 1.5 T scanner (n = 43; MAGNETOM Avanto fit, Siemens Healthcare, Erlangen, Germany) or a 3.0 T scanner (n = 14; Ingenia 3.0T, Philips Healthcare, Best, the Netherlands), both with a slice thickness of 5 mm.

## Posttreatment imaging assessment

Posttreatment recanalization rate was scored as the extended Thrombolysis in Cerebral Infarction (eTICI) score by an independent core lab from the Collaboration for New Treatments of Acute Stroke (CONTRAST) consortium (n = 40). For patients not included in one of the CONTRAST trials or for whom there was no core lab observation available (n = 19), posttreatment DSAs were evaluated by an independent, blinded observer (>5 years of experience) who is part of the CONTRAST consortium core lab.

## CTP data post-processing

CTP data were processed using Siemens syngo.via CT Neuro Perfusion (version VB40; Siemens Healthcare, Erlangen, Germany). The software was used in a research environment in order to automatically produce the results and to be able to modify the parameters and thresholds. CTP data were checked for severe patient motion, arterial input function curve, and presence of metal artifacts. Four core estimation approaches were investigated: Approach 1 represents the CBV-based core estimation approach (threshold CBV<1.2mL/100mL) with additional smoothing filter [8]. Approach 2 represents the conventional, CBV-based core estimation approach (threshold CBV<1.2mL/100mL) without smoothing filter, approach 3 represents a rCBF-based approach with thresholds derived from another commercial package (RAPID; iSchemaView) (threshold rCBF<30% + smoothing filter), and approach 4 represents the rCBF-based core estimation approach from syngo.via used for research purposes (threshold rCBF<20% + smoothing filter). We chose these specific four approaches since these are commonly used in daily clinical practice or recommended in the current guidelines [6, 7].

### Data co-registration and follow-up imaging assessment

The follow-up DWI images (median 23h) were registered (rigid registration) to the baseline CTP using Elastix [15]. Results were visually inspected to assure correct alignment (JWH). The follow-up infarct volumes were segmented on DWI using a semi-automated segmentation method [16] with subsequent visual assessment by an expert neuroradiologist with >15 years of experience (CBLMM) who was blinded to all clinical information but occlusion side. If necessary, the automated segmentation results were manually adjusted after visual expert assessment using ITK-S-NAP [17]. DWI images were screened for scattered lesions. We defined a scattered lesion as at least two separate hyperintense DWI lesions within the territory of one of the major cerebral arteries [18]. DWI scattered lesion patterns were scored as absent (no hyperintense lesions), mild (>0 but <5 hyperintense lesions), moderate (≥5 but <10 separate hyperintense lesions), or severe (≥10 separate hyperintense lesions). All hyperintense lesions on DWI were included for both volumetric and spatial analysis. DWI was chosen for follow-up infarct segmentation as it shows good agreement with the follow-up infarct volume and it is more sensitive than FLAIR for the detection of acute ischemic stroke lesions [19, 20]. The ADC maps were consulted to prevent including T2 shine-through lesions in the DWI lesion. For spatial agreement analysis, the CTP-estimated ischemic core segmentation was co-registered to the baseline CTP.

### Assessment of volumetric and spatial agreement and other statistical analyses

We calculated the volume difference between the estimated CTP ischemic core and DWI follow-up infarct volume as

$$Volume_{difference} = Volume_{DWI} - Volume_{CTP.} \tag{1}$$

Negative values indicate overestimation of the ischemic core by the CTP software. The intraclass correlation coefficient (ICC) estimates and their 95% confidence intervals were calculated to assess the agreement between the estimated CTP ischemic core volume for each approach and the follow-up DWI infarct volume. We based the ICC estimates on a mean-rating (k = 2), absolute agreement, 2-way mixed-effects model and classified the degree of agreement as previously suggested (ICC <0.5 = poor agreement, ICC 0.5–0.75 = moderate agreement, ICC 0.75–0.9 = good agreement, and ICC >0.9 = excellent agreement) [21]. Bland-Altman analyses were performed to compare the estimated CTP ischemic core and follow-up DWI infarct volumes. Proportional bias was assessed using linear regression. The spatial overlap between the CTP and DWI segmentation was calculated using FSLMaths [22]. We calculated the Dice similarity coefficient using the 'fslr' package in R to quantify the spatial agreement between the CTP-estimated ischemic core and follow-up DWI lesion for the four core estimation approaches [23].

We reported the frequency and summary statistics for ordinal and continuous baseline characteristics. We performed linear regression analyses to determine the association between the time from imaging to reperfusion and spatial and volume difference. To determine if there were statistically significant differences in volumetric or spatial agreement between the estimated CTP ischemic core and the follow-up DWI lesion for the four approaches, we performed Friedman tests. Finally, we performed a sensitivity analysis of the volumetric and spatial agreement between CTP ischemic core and 24h follow-up DWI infarct lesion in patients with incomplete (eTICI 2b) vs. complete reperfusion (eTICI 3) to determine the degree of differences between subgroups in our analysis. All statistical analyses were performed using R (R Core Team (2020). R: A language and environment for statistical computing, R Foundation for Statistical Computing, Vienna, Austria. https://www.R-project.org).

### Ethics statement

This study was reviewed by the Medical Ethical Committee Board of the Amsterdam University Medical Centers (location AMC) and informed consent was waived (Reference W19_281#19.334) as retrospective, observational studies do not fall under the scope Medical Research Involving Human Subject Act (WMO).

## Results

Three hundred one patients were presented to our comprehensive stroke center for EVT between November 2017 and October 2020 and received baseline CTP imaging. For 284/301 patients, baseline CTA imaging showed an anterior circulation large vessel occlusion. Eighty-four (30%) patients received follow-up DWI at median 23h (IQR 18–34) after CTP. Most patients (49/59; 83%) in our study cohort were included in one of the randomized controlled trials of the CONTRAST consortium (i.e., MR CLEAN-NO IV, MR CLEAN-MED or MR CLEAN-LATE) [24] and received 24h follow-up DWI as part of the pre-specified follow-up imaging of the concerning trial [25–27]. After application of the exclusion criteria, we included a total of 59 patients in our analysis (Fig 1).

Table 1 shows a detailed description of the baseline characteristics. Median age was 71 (IQR 58–76) years. Median National Institutes of Health Stroke Scale Score (NIHSS) was 15 (IQR 9–18) and most patients presented within 6 hours after symptom onset (55/59; 93%). Compared to the overall Dutch stroke population described in the MR CLEAN Registry [28], more patients were female in our cohort (59% vs. 47%) and less patients received IV alteplase (49% vs. 78%).

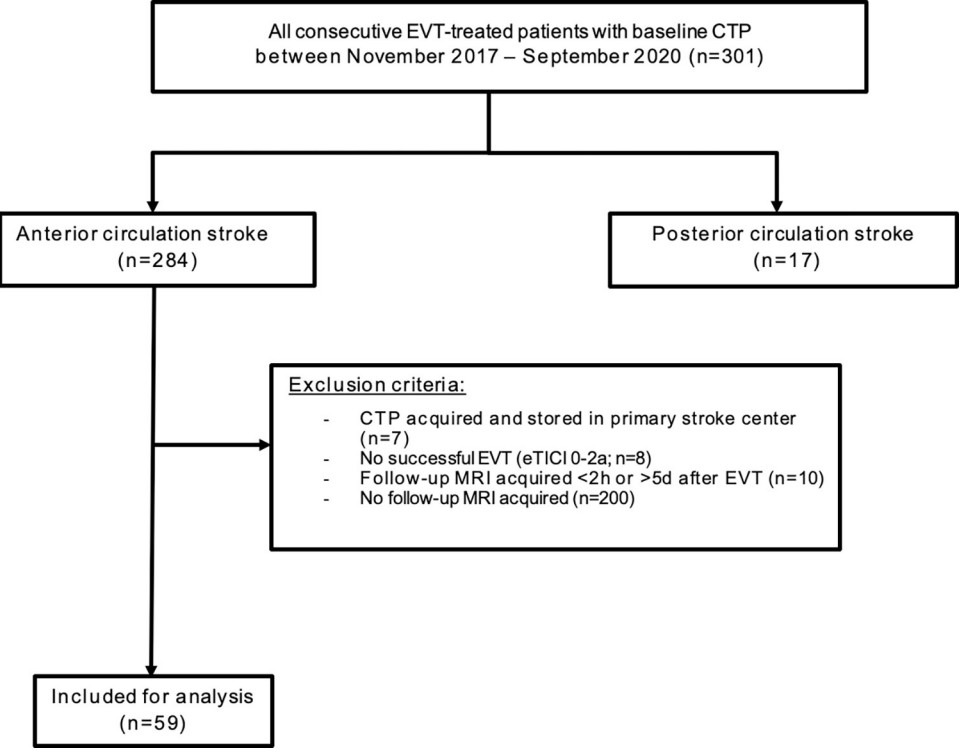

**Fig 1. Flowchart of patient selection.**

**Table 1. Baseline characteristics.**

| | Study cohort (n = 59) |
|---|---|
| **Clinical characteristics** | |
| **Age (yr)**–median (IQR) | 71 (58–77) |
| **Female**–n (%) | 36 (59) |
| **NIHSS score**–median (IQR) [known in] | 15 (9–19) [n = 57] |
| **IVT administered**–n (%) | 29 (49) |
| **Onset-to-imaging time (min)**–median (IQR) [known in] | 83 (58–188) [n = 57] |
| **Imaging-to-reperfusion time (min)**–median (IQR) [known in] | 83 (63–114) [n = 58] |
| **Onset-to-groin time (min)**–median (IQR) [known in] | 140 (105–222) [n = 53] |
| **Imaging characteristics** | |
| **Occlusion location on baseline CTA**–n (%) | |
| **Intracranial ICA** | 3 (5) |
| **ICA-T** | 11 (19) |
| **M1** | 42 (70) |
| **M2** | 3 (5) |
| **Collateral status**–n (%) [known in] | [n = 58] |
| **0** | 2 (3) |
| **1** | 18 (31) |
| **2** | 23 (39) |
| **3** | 15 (25) |
| **Median baseline ischemic core volume on CTP (mL); approach 1** –median (IQR) | 15 (4–31) |
| **Median baseline ischemic core volume on CTP (mL); approach 2** –median (IQR) | 27 (11–53) |
| **Median baseline ischemic core volume on CTP (mL); approach 3** –median (IQR) | 39 (20–95) |
| **Median baseline ischemic core volume on CTP (mL); approach 4** –median (IQR) | 11 (3–31) |
| **Posttreatment recanalization rate (eTICI)** | |
| **0** | 1 (2) |
| **2a** | 3 (5) |
| **2b** | 20 (34) |
| **2c** | 8 (14) |
| **3** | 27 (46) |
| **Follow-up DWI infarct volume (mL)**–median (IQR) | 11 (5–42) |
| **Median time between baseline CTP and follow-up DWI (hrs)**–median (IQR) | 23 (18–34) |

ASPECTS = Alberta Stroke Program Early CT Score; CTP = CT perfusion; ICA = intracranial carotid artery; ICA-T = intracranial carotid artery terminus; IVT = IV alteplase; IQR = interquartile range; NIHSS = National Institute of Health Stroke Scale. If the [known in] number is not shown, the variable was known in all patients.

All other clinical and imaging characteristics were comparable to the population from the MR CLEAN Registry [28].

## Volumetric agreement analysis

The median volume difference between the follow-up DWI infarct and the CTP ischemic core per estimation approach was: approach 1: 0 (IQR -10 to 18) mL, approach 2: -8 (IQR -23 to 13) mL, approach 3: -16 (IQR -44 to 2) mL, and approach 4: 2 (IQR -6 to 20) mL. Most patients showed a moderate (14/59; 24%) or severe (28/59; 34%) scattered lesion pattern on the follow-up DWI. The CTP ischemic core volumes were significantly different ($p<0.01$) for all combinations of core estimation approaches, except for the combination approach 1 vs. approach 4

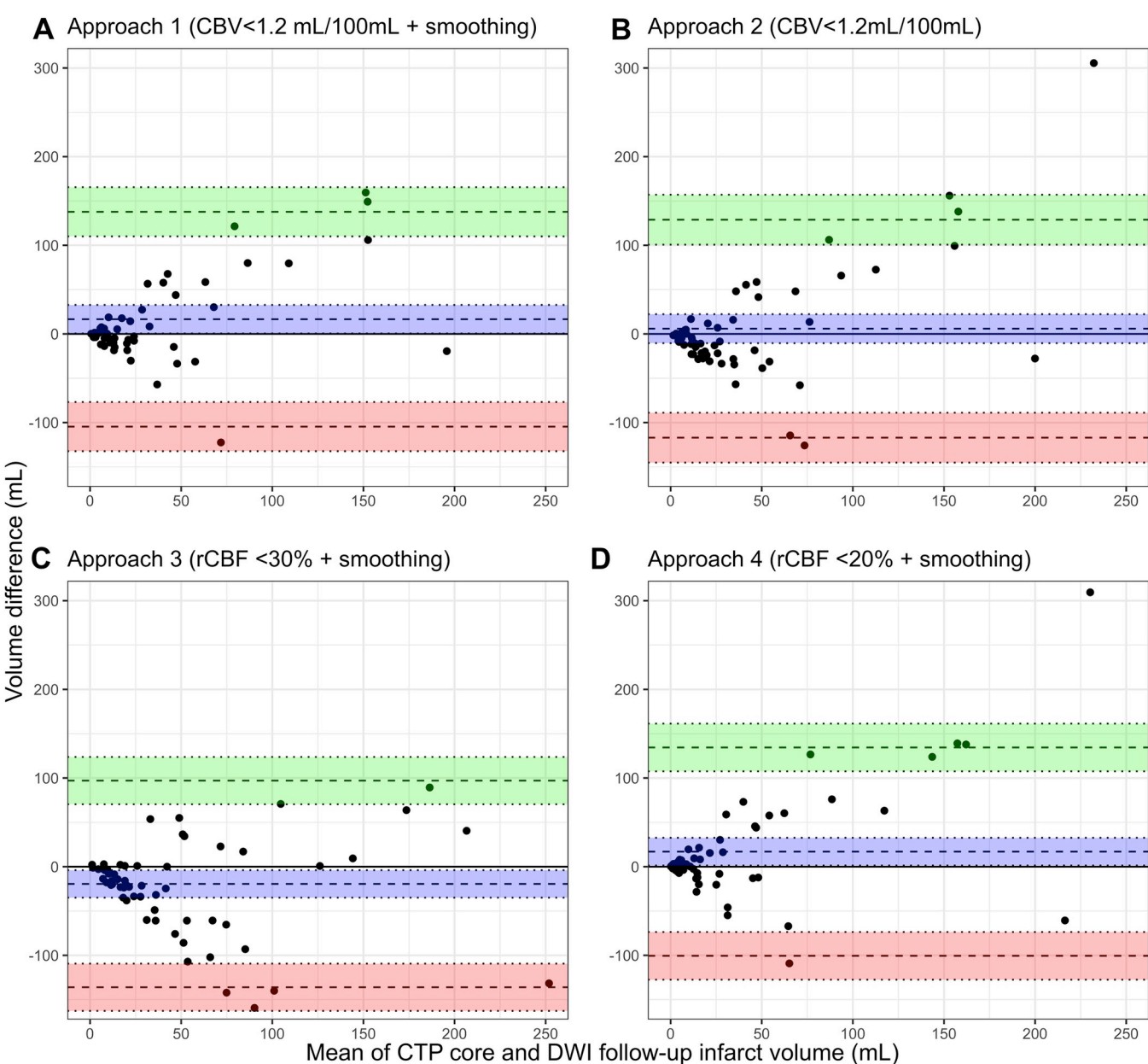

**Fig 2.** Bland-Altman plots comparing the estimated CTP ischemic core volume and DWI follow-up infarct volume for (a) approach 1, (b) approach 2, (c) approach 3, and (d) approach 4. The mean bias (blue), lower (red) and upper (green) Limits of Agreement are shown with 95% confidence intervals. The bias with 95% confidence intervals is shown in blue. Negative values indicate overestimation by CTP. CTP = CT perfusion; DWI = diffusion weighted imaging.

($p$ = 0.4). Of note, for individual cases, we still found volume differences up to 40 mL between approach 1 and approach 4 (Fig 2). ICC estimates showed moderate–good volumetric agreement for all core estimation approaches (approach 1: ICC 0.61[95% CI 0.39–0.75], approach 2: ICC 0.61[95% CI 0.40–0.75], approach 3: ICC 0.76[95% CI 0.64–0.85], approach 4: ICC 0.67 [95% CI 0.48–0.78]).

The Bland-Altman plots are shown in Fig 2. Linear regression showed that proportional bias was present for 3 of the 4 core estimation approaches (approach 1: p<0.01, approach 2: p<0.01, and approach 4: p<0.01).

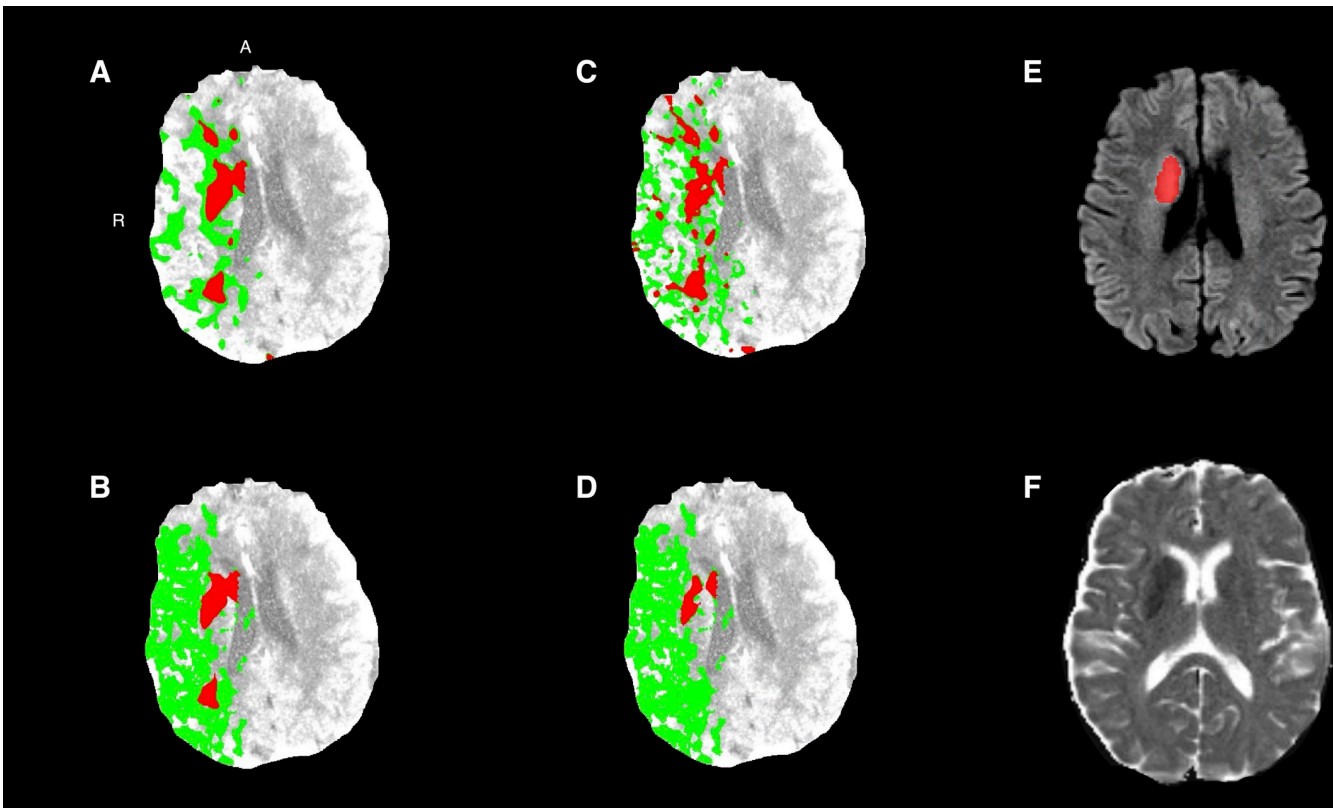

**Fig 3. Baseline CTP of a patient with a right-sided M1 occlusion with successful reperfusion (eTICI 3).** The ischemic core (red) and penumbra (green) for (a) approach 1, (b) approach 2, (c) approach 3, and (d) approach 4. (e) Follow-up DWI acquired at 17 hours after baseline imaging and (f) follow-up DWI image with follow-up infarct segmentation (red). A = anterior; CTP = CT perfusion; DWI = diffusion weighted imaging; MRI = magnetic resonance imaging; R = right.

CTP ischemic core overestimation of >10 mL was not uncommon (approach 1:14/59 (24%), approach 2: 26/59 (48%), approach 3: 33/59 (56%), approach 4: 12/59 (20%). Severe volume overestimation >50 mL by CTP occurred significantly more often for approach 3 (24%, 14/59) compared to approach 1 (3% 2/59; $p<0.01$), approach 2 (7%, 4/59; $p = 0.01$), and approach 4 (7%, 4/59; $p = 0.01$). Please see S1 Fig for a schematic overview of the CTP ischemic core volumes per core estimation approach. Please see S2 Fig for the scatter plots of the volumetric agreement between the estimated CTP core and the 24h follow-up DWI infarct.

Fig 3 shows examples of CTP ischemic core estimates (red) for the four estimation approaches and follow-up DWI and ADC images.

## Spatial agreement analysis

The median Dice was low for all core estimation approaches (approach 1: 0.16[IQR 0.02–0.31], approach 2: 0.15[IQR 0.02–0.30], approach 3: 0.21[IQR 0.06–0.35], approach 4: 0.15 [IQR 0.01–0.32]). See S3 Fig for more details.

## Effect of imaging-to-reperfusion time on the spatial and volumetric accuracy

The median time between CTP acquisition and reperfusion was 83 (IQR 63–114) minutes. Longer time between imaging and reperfusion was not associated with spatial accuracy, but

**Table 2. Volumetric agreement for patients with eTICI 2b vs. eTICI 3.**

|  | Approach 1 | Approach 2 | Approach 3 | Approach 4 |
|---|---|---|---|---|
| **ICC eTICI 2b (n = 20)** | 0.62 (95% CI 0.18–0.83) | 0.62 (95% CI 0.18–0.83) | 0.78 (95% CI 0.53–0.90) | 0.69 (95% CI 0.33–0.86) |
| **ICC eTICI 3 (n = 27)** | 0.65 (95% CI 0.33–0.82) | 0.68 (95% CI 0.38–0.83) | 0.80 (95% CI 0.62–0.90) | 0.69 (95% CI 0.41–0.84) |

was associated with increased volume difference for all approaches (range 0.9–1.0 mL per minute). Scatter plots are shown in S4 and S5 Figs.

### Sensitivity analysis comparing patients with incomplete vs. complete reperfusion (eTICI 2b vs. eTICI 3)

For patients with incomplete (eTICI 2b; n = 20) vs. complete reperfusion (eTICI 3; n = 27), median CTP ischemic core volumes were as follows: approach 1: 12 (IQR 4–26) mL vs. 16 (IQR 11–34) mL, approach 2: 18 (IQR 11–41) mL vs. 31 (IQR 21–55) mL, approach 3: 34 (IQR 16–76) mL vs. 42 (IQR 24–112) mL, and approach 4: 7 (IQR 2–24) mL vs. 12 (IQR 5–33) mL.

The ICC estimates for both patients with eTICI 2b and eTICI 3 ranged from moderate-good (Table 2).

The median Dice indicated low spatial agreement for all estimation approaches for both patients with eTICI 2b and eTICI3 (Table 3).

Median times between CTP and FU DWI for patients who achieved non-complete (eTICI 2b) reperfusion and complete (eTICI 3) reperfusion were 21 (IQR 19–33) and 23 (IQR 17–31) hours, respectively ($p = 0.8$). Median follow-up DWI volumes were 12 (IQR 4–22) mL and 16 (IQR 11–31) mL for the eTICI 2b and eTICI 3 subgroup, respectively ($p = 0.3$)."

## Discussion

Our study showed good–excellent volumetric agreement between the CTP ischemic core and the follow-up DWI lesion for four core estimation approaches for EVT-treated patients with complete reperfusion. The core estimation approach based on CBV<1.2 mL/100 mL with smoothing filter showed the best volumetric agreement. Overall, the spatial agreement was low (Dice range: 0.16–0.21). We found the highest spatial accuracy for the core estimation approach based on rCBF <30% with smoothing filter. If we also included patients with successful–yet not complete–reperfusion, we found moderate spatial and volumetric agreement. Volumetric overestimation >10mL was not uncommon and occurred in 20–56% of the successfully reperfused patients depending on the core estimation approach used. For patients with volumetric overestimation of approximately 10mL it is not likely that this would have affected the treatment decision for these patients. For 14/59 (24%) patients, the rCBF-based core estimation approach resulted in severe volumetric overestimation >50 mL whereas severe volumetric overestimation occurred in 2/59 (3%) patients using the core estimation approach based on CBV <1.2 mL/100mL with smoothing filter. In contrast to volumetric overestimation of approximately 10mL, it is possible that this could have resulted in falsely withholding these patients from EVT from. Our results suggest that severe volumetric overestimation was

**Table 3. Spatial agreement for patients with eTICI 2b vs. eTICI 3.**

|  | Approach 1 | Approach 2 | Approach 3 | Approach 4 |
|---|---|---|---|---|
| **Dice TICI 2b (n = 20)–median (IQR)** | 0.16 (IQR 0.01–0.33) | 0.13 (IQR 0.01–0.32) | 0.16 (IQR 0.06–0.35) | 0.05 (IQR 0.00–0.32) |
| **Dice TICI 3 (n = 27)–median (IQR)** | 0.19 (IQR 0.07–0.29) | 0.22 (IQR 0.19–0.28) | 0.26 (IQR 0.18–0.36) | 0.18 (IQR 0.07–0.33) |

associated with the occlusion location if the ischemic core approach was based on rCBF with a threshold of rCBF <30% or rCBF <20% (approach 3 and approach 4).

A recent study compared baseline estimated ischemic core volumes of syngo.via to estimated ischemic core volumes from RAPID [12]. They found similar results between syngo.via (version VB30) and RAPID if rCBF <20% was used for ischemic core estimation by syngo.via. However, since the current recommended core estimation approach of syngo.via is not based on rCBF, these results are not generalizable to a clinical setting where a recommended CBV-based approach is used. Also, this study did not compare the CTP results to follow-up DWI, in contrast to the present study. Another study compared the core estimates from four software packages (i.e., RAPID, VEOcore, syngo.via, and Olea) and found volume differences up to 33 mL between the different software packages [11].

Other studies on CTP accuracy found median volume differences between the CTP ischemic core volume and follow-up imaging of 30 mL and 13 mL [29, 30]. These studies, however, focused on different CTP software packages (i.e., IntelliSpace Portal and RAPID) with different ischemic core parameters and thresholds (relative mean transit time (rMTT) ≥145% + CBV <2.0 mL/100mL and rCBF <30%), which hampers the comparison of these results to our findings.

The spatial agreement between the estimated CTP ischemic core and follow-up DWI infarct seems poor. However, these results are in line with a previous comparison between RAPID CTP estimated ischemic core and 24h follow-up DWI infarct volume [31]. Comparing Dice scores with the aim to determine the best clinical performance should be performed with caution as the Dice score is very easily negatively affected by 'false negative voxels'–especially in relatively small segmentations, such as infarct segmentations [32]. This would result in a higher Dice for a core estimation approach with more frequent CTP ischemic core overestimation (e.g., for approach 3 in our analysis). Yet, this approach would not be optimal for clinical decision making as overestimation could lead to falsely withholding patients from EVT.

The moderate volumetric agreement (ICC) in our study could be a result of infarct growth in case of incomplete (micro- or macrovascular) reperfusion or delay between imaging and reperfusion. The significant association between longer imaging-to-reperfusion time and larger volume difference for all core estimation approaches supports this hypothesis. Moreover, if only patients with complete reperfusion were taken into account, we found good–excellent volumetric accuracy for all core estimation approaches.

The volumetric agreement (ICC) outperformed the spatial agreement for all ischemic core estimation approaches. It is plausible that scattered subcortical white matter lesions–which are likely to be a result of distal embolization during the EVT procedure–contribute to this finding. These lesions do not have a major impact on the volumetric agreement, but do affect the degree of 'false negative voxels' and thus the spatial agreement between the CTP ischemic core and follow-up infarct as mentioned earlier.

Several limitations to our study should be noted. First, all included patients who underwent EVT had relatively small ischemic core volumes (median range 13–40 mL). More specifically, 43/59 (73%) patients received baseline imaging in the hyperacute time window (i.e., within 3 hours after symptom onset) where CTP is not recommended for selection for EVT. The median onset-to-imaging and onset-to-groin times were 83 and 140 minutes, respectively. Therefore, we are not able to draw conclusions about the volumetric or spatial accuracy of syngo.via for large core volumes >70 mL or for patients who presented outside hyperacute time window, for example due to transfer from a mothership hospital. Of note, for patients who present within this time window, CTP is not recommended for selection for EVT. Second, we only included patients who received follow-up DWI which could have introduced selection bias. In the Netherlands, follow-up DWI is generally only acquired for research purposes and

therefore sparsely performed in clinical practice. However, from a local cohort of 291 patients who received EVT outside one of the Collaboration for New Treatments of Acute Stroke (CONTRAST) trials, the median CTP-estimated core volume was 11 (IQR 5–32) mL, so it is unlikely that only patients with smaller core volumes were selected in our current study population. Third, we compared the estimated CTP ischemic core volume at baseline with the 24h follow-up DWI infarct volume in patients with successful reperfusion as previously suggested [33]. Although DWI accurately differentiates between cytotoxic and vasogenic edema [34] and is commonly used to determine the follow-up infarct volume, there is no ideal reference standard to determine the infarct volume [33]. Fourth, the infarct is likely to expand in the time between CTP acquisition and reperfusion–especially in patients with incomplete micro- or macrovascular reperfusion (i.e., eTICI 2b or 2c)–, which make the degree of reperfusion and timing of the follow-up imaging important factors to consider when performing accuracy assessments [35]. Also, co-registration between CTP and DWI is not optimal and could have negatively influenced our findings. To minimize this effect, we visually inspected all registration results. Previous studies have assessed this by determining the ventricle overlap between CTP-DWI and found suboptimal spatial agreement (Dice 0.8) [25, 27]. Fifth, we included patients with both 1.5 and 3.0 T follow-up DWI scans. As it has been shown that there might be great volumetric and spatial differences between the two field strengths, this could have influenced our results [36]. Finally, we were not able to assess the accuracy of the penumbra estimations since patients who did not undergo EVT were not included in our cohort.

## Conclusions

In successfully reperfused patients who underwent EVT, syngo.via CTP ischemic core estimation showed moderate volumetric and spatial agreement with the follow-up infarct on DWI. For patients with complete reperfusion after EVT, the volumetric agreement was excellent. A CTP core estimation approach based on CBV<1.2 mL/100mL with smoothing filter least often overestimated the follow-up infarct volume and is therefore preferred for clinical decision making using syngo.via.

## Supporting information

**S1 Fig. Boxplots show the distribution of CTP ischemic core volume per estimation approach.** The estimated CTP ischemic core volume was statistically significantly different for the four core estimation approaches using Friedman test, $\chi^2$ = 102.31, p<0.001. Pairwise Wilcoxon signed rank test between groups revealed statistically significant differences in CTP ischemic core volume between approach 1-approach 2 (p<0.001), approach 1-approach 3 (p<0.001), approach 2-approach 3 (p<0.001), approach 2-approach 4 (p<0.001), and approach 3-approach 4 (p<0.001). CTP = computed tomography perfusion.
(TIF)

**S2 Fig.** Scatter plots show the agreement between the estimated CTP ischemic core volume and the follow-up DWI infarct lesion for (A) approach 1, (B) approach 2, (C) approach 3, and (D) approach 4. The solid grey line represents the identity line. Points below the identity line (grey) indicate a larger CTP ischemic core volume compared to the follow-up DWI lesion, i.e., overestimation by CTP. Points above the identity indicate underestimation by CTP or infarct growth. CTA = CT angiography; CTP = CT perfusion; DWI = diffusion weighted imaging; ICA = intracranial carotid artery; ICA-T = intracranial carotid artery terminus; M1 = M1 (horizontal) segment of the middle cerebral artery; M2 = M2 (insular) segment of the middle

cerebral artery.
(TIF)

**S3 Fig. Boxplots show the distribution of the Dice similarity coefficient (Dice) per estimation approach.** Dice was statistically significantly different for the different approaches using Friedman test, $\chi^2$ = 27.45, p<0.001. Pairwise Wilcoxon signed rank test between groups revealed statistically significant differences in Dice between approach 1-approach 3 (p = 0.005), approach 2-approach 3 (p = 0.001), and approach 3-approach 4 (p = 0.001). CTP = computed tomography perfusion.
(TIF)

**S4 Fig.** Scatter plots show the association between time from CTP imaging to reperfusion and Dice similarity coefficient for (a) approach 1, (b) approach 2, (c) approach 3, and (d) approach 4. *R* = Pearson correlation coefficient.
(TIF)

**S5 Fig.** Scatter plots show the association between time from CTP imaging to reperfusion and volume difference for (a) approach 1, (b) approach 2, (c) approach 3, and (d) approach 4. *R* = Pearson correlation coefficient.
(TIF)

## Author Contributions

**Conceptualization:** Jan W. Hoving, Miou S. Koopman, Henk A. Marquering, Bart J. Emmer, Charles B. L. M. Majoie.

**Data curation:** Jan W. Hoving, Henk van Voorst.

**Formal analysis:** Jan W. Hoving, Henk A. Marquering.

**Investigation:** Jan W. Hoving.

**Methodology:** Jan W. Hoving, Henk A. Marquering.

**Resources:** Jan W. Hoving.

**Software:** Jan W. Hoving, Marcus Brehm.

**Supervision:** Henk A. Marquering, Bart J. Emmer, Charles B. L. M. Majoie.

**Visualization:** Jan W. Hoving.

**Writing – original draft:** Jan W. Hoving.

**Writing – review & editing:** Jan W. Hoving, Miou S. Koopman, Manon L. Tolhuisen, Henk van Voorst, Marcus Brehm, Olvert A. Berkhemer, Jonathan M. Coutinho, Ludo F. M. Beenen, Henk A. Marquering, Bart J. Emmer, Charles B. L. M. Majoie.

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
