## [Decision Letter · Decision Letter 0]

23 Feb 2022

PONE-D-22-00274Accuracy of CT perfusion ischemic core volume and location estimation: a comparison between four ischemic core estimation approaches using syngo.viaPLOS ONE

Dear Dr. Hoving,

Thank you for submitting your manuscript to PLOS ONE. After careful consideration, we feel that it has merit but does not fully meet PLOS ONE’s publication criteria as it currently stands. Therefore, we invite you to submit a revised version of the manuscript that addresses the points raised during the review process.

We look forward to receiving your revised manuscript.

Kind regards,

Marios-Nikos Psychogios

Academic Editor

PLOS ONE

Journal Requirements:

"HAM is co-founder and shareholder of Nicolab. All other authors report no conflicts of interest. BJE reports grants from Stryker Neurovascular and personal fees from Dekra and from Novartis, outside the submitted work. CBLMM reports grants from TWIN, during the conduct of the study and grants from CVON/Dutch Heart Foundation and from Stryker outside the submitted work (paid to institution) and is shareholder of Nicolab. All other authors did not receive support from any organization for the submitted work, had no financial relationships with any organizations that might have an interest in the submitted work in the previous three years, and had no other relationships or activities that could appear to have influenced the submitted work."

We note that you received funding from a commercial source: Stryker Neurovascular, Dekra, Novartis, TWIN, CVON/Dutch Heart Foundation and Stryker

Reviewers' comments:

Reviewer's Responses to Questions

**Comments to the Author**

1. Is the manuscript technically sound, and do the data support the conclusions?

Reviewer #1: Yes

Reviewer #2: Yes

2. Has the statistical analysis been performed appropriately and rigorously? 

Reviewer #1: Yes

Reviewer #2: Yes

3. Have the authors made all data underlying the findings in their manuscript fully available?

Reviewer #1: Yes

Reviewer #2: No

4. Is the manuscript presented in an intelligible fashion and written in standard English?

Reviewer #1: Yes

Reviewer #2: Yes

5. Review Comments to the Author

Reviewer #1: This is a well written study, which addresses an important clinical question with merging importance especially, as LVOs are more and more prone to treatment in expanded time windows and different approaches for estimation of penumbra and core assessment are used in the field.

This already leads to one of my major concerns regarding this work:

- the authors chose very early treatment time points (Mr Clean onset -door: 240min, here 140min; Hermes Onset to random 196min vs 83 onset to image here). Firstly these are timelines which in my eyes do not apply to most clinical settings, as the o-d times are much longer, especially for shipped patients. Secondly the perfusion approach is not applicable to the timelines studied here, as current guideline recommendation is beyond the 4,5 rep. 6hrs window. Thus it would have made more sense to focus on later time points.

-Another point is the delay of the MRI , which dependent on TICI success will influence the final DWI lesion size. can the authors comment on that and provide an analysis of 2b and final DWI size compared to 3?

- another drawback is the use of different MRI field strengths within the cohort (3 vs 1,5T), there is some evidence form stereotactic studies, that there might be great differences between the to field strengths (doi: 10.3340/jkns.2014.55.3.136). Can the authors please comment on that?

- Another problem could be the non blinded TICI evaluation , which was done by the interventionalist and might lead to BIAS, It would be worthwhile to crosscheck those by a blinded rater.

- Table 1 shows, that for some metrics not all informations form every patient were available....how did the authors deals with that?

Minor: Table 1:

median follow up DWI time 23min? probably 23hrs...?

Line 307 duplication of "from"

Suppl. figure S1 in my eyes does not add visual information, as I can visually barely detect sign. differences between the approaches.

Reviewer #2: Overall interesting topic which needs further research as these thresholds are still to be validated and fu infarct volume is a good approximation.

One reference you might add to the discussion as it looks at a similar topic outlining the discrepancies between different software packages:

https://pubmed.ncbi.nlm.nih.gov/33216157/

6. PLOS authors have the option to publish the peer review history of their article (what does this mean?). If published, this will include your full peer review and any attached files.

Reviewer #1: No

Reviewer #2: No

---

## [Author Response · Author response to Decision Letter 0]

24 Mar 2022

Reviewer #1: 

This is a well written study, which addresses an important clinical question with merging importance especially, as LVOs are more and more prone to treatment in expanded time windows and different approaches for estimation of penumbra and core assessment are used in the field.

1.1 This already leads to one of my major concerns regarding this work:

- the authors chose very early treatment time points (Mr Clean onset -door: 240min, here 140min; Hermes Onset to random 196min vs 83 onset to image here). Firstly, these are timelines which in my eyes do not apply to most clinical settings, as the o-d times are much longer, especially for shipped patients. 

We thank the reviewer for pointing us to this limitation. We agree that the patients in our study cohort received imaging in the hyperacute time window and relatively shortly after symptom onset (median 83 minutes). This could partially be explained by the fact that most patients included in our analysis were directly admitted to the EVT-capable center and did not have to be transferred from a primary stroke center. Another reason could be that most patients who were included in the MR CLEAN-NO IV trial received follow-up DWI imaging and were subsequently included in our analysis. The MR CLEAN-NO IV trial studied the added value of IV alteplase prior to EVT in the 0-4.5h time window.

We have added this as a limitation to our study. (Page 20, line 557-564): 

“Several limitations to our study should be noted. First, all included patients who underwent EVT had relatively small ischemic core volumes (median range 13-40 mL). More specifically, 43/59 (73%) patients received baseline imaging in the hyperacute time window (i.e., within 3 hours after symptom onset) where CTP is not recommended for selection for EVT. The median onset-to-imaging and onset-to-groin times were 83 and 140 minutes, respectively. Therefore, we are not able to draw conclusions about the volumetric or spatial accuracy of syngo.via for large core volumes >70 mL or for patients who presented outside hyperacute time window, for example due to transfer from a mothership hospital.”

1.2 Secondly, the perfusion approach is not applicable to the timelines studied here, as current guideline recommendation is beyond the 4,5 rep. 6hrs window. Thus it would have made more sense to focus on later time points.

We thank the reviewer for this comment and we acknowledge that CTP is indeed not recommended for selection for IVT or EVT in the 0-4.5, resp. 0-6h time windows. We agree that CTP should currently not be used for patient selection for EVT in the earlier time windows. However, we feel that for diagnostic purposes, it is relevant to quantify the spatial and volumetric accuracy, both for the hyperacute as in the later time windows. We agree that it would have been interesting to include more patients in the later time window. However, we were only able to include patients with follow-up DWI. These follow-up DWI data were mostly acquired in the setting of either the MR CLEAN-NO IV or the MR CLEAN-MED trials which were focusing on patients in the 0-4.5 or 0-6h time window.

We have specified the origin of our study cohort in the Results section and specified that patients received 24h follow-up imaging as part of follow-up imaging from the concerning randomized trial. (Page 11, line 275-279): 

“Most patients (49/59; 83%) in our study cohort were included in one of the randomized controlled trials of the CONTRAST consortium (i.e., MR CLEAN-NO IV, MR CLEAN-MED or MR CLEAN-LATE) [24] and received 24h follow-up DWI as part of the pre-specified follow-up imaging of the concerning trial [25–27].”

We have addressed the Discussion section (Page 20, line 558-565):

“More specifically, 43/59 (73%) patients received baseline imaging in the hyperacute time window (i.e., within 3 hours after symptom onset) where CTP is not recommended for selection for EVT. The median onset-to-imaging and onset-to-groin times were 83 and 140 minutes, respectively. Therefore, we are not able to draw conclusions about the volumetric or spatial accuracy of syngo.via for large core volumes >70 mL or for patients who presented outside hyperacute time window, for example due to transfer from a mothership hospital. Of note, for patients who present within this time window, CTP is not recommended for selection for EVT.”

1.3 Another point is the delay of the MRI, which dependent on TICI success will influence the final DWI lesion size. can the authors comment on that and provide an analysis of 2b and final DWI size compared to 3?

We fully agree with the reviewer and acknowledge that this is a major limitation of our analysis. 

We have emphasized this in the limitations section of our Discussion (Page 21, lines 589-593): 

“Fourth, the infarct is likely to expand in the time between CTP acquisition and reperfusion – especially in patients with incomplete micro- or macrovascular reperfusion (i.e., eTICI 2b or 2c) –, which make the degree of reperfusion and timing of the follow-up imaging important factors to consider when performing accuracy assessments [35].”

Also, we have added an additional sensitivity analysis (“Sensitivity analysis comparing patients with incomplete vs. complete reperfusion (eTICI 2b vs. eTICI 3”) between the (newly scored) eTICI 2b vs. eTICI 3 subgroups to the Results section. (Page 16-17, lines 399-481): 

We have performed additional analyses on the volumetric (ICC) and spatial (Dice) agreement for these two subgroups and found that for patients with complete reperfusion, all approaches showed moderate-good volumetric agreement and low spatial agreement between the estimated CTP core the 24h follow-up DWI infarct. (Page 16-17, Tables 2-3)

1.4 Another drawback is the use of different MRI field strengths within the cohort (3 vs 1,5T), there is some evidence form stereotactic studies, that there might be great differences between the two field strengths (doi:10.3340/jkns.2014.55.3.136). Can the authors please comment on that?

We thank the reviewer for pointing us to this important issue. We have added this as a limitation to our study in the Discussion section of our manuscript. (Page 21, lines 594-597):

“Fifth, we included patients with both 1.5 and 3.0 T follow-up DWI scans. As it has been shown that there might be great volumetric and spatial differences between the two field strengths, this could have influenced our results [36].”

1.5 Another problem could be the non-blinded TICI evaluation, which was done by the interventionalist and might lead to BIAS. It would be worthwhile to crosscheck those by a blinded rater.

We agree that the TICI evaluation by the interventionalist could have introduced bias. Unfortunately, there were no independent core lab observations available at the time of the writing of the manuscript. However, in the meantime, the data of the MR CLEAN-NO IV trials has become available. As most patients included in our analysis were also included in the MR CLEAN-NO IV trial, we could resort to the independent core lab TICI evaluations. These observations were all scored blindly by an independent observer who was part of the Collaborations for New Treatment of Acute Stroke (CONTRAST) consortium core lab. For patients who were not included in the MR CLEAN-NO IV trial or for whom there was no core lab observation available (n=19), the post-EVT DSA images were crosschecked by an independent, blinded rater who is part of the CONTRAST core lab.

We have adapted the specification of the posttreatment imaging assessment this in our manuscript. (Page 7, lines 151-156):

“Posttreatment recanalization rate was scored as the extended Thrombolysis in Cerebral Infarction (eTICI) score by an independent core lab from the Collaboration for New Treatments of Acute Stroke (CONTRAST) consortium (n=40). For patients not included in one of the CONTRAST trials or for whom there was no core lab observation available (n=19), posttreatment DSAs were evaluated by an independent, blinded observer (>5 years of experience) who is part of the CONTRAST consortium core lab.”

We have adapted S2 Fig with the updated eTICI scores. (S2 Fig)

1.6 Table 1 shows, that for some metrics not all informations form every patient were available....how did the authors deals with that?

We thank the reviewer for noticing. Indeed, unfortunately, not all clinical baseline variables were available for all patients at the time of writing of the manuscript. However, in the meantime, clinical data from the MR CLEAN-NO IV trial has become available. Since a considerable number of patients included in our analysis were also included in the MR CLEAN-NO IV trial, we were able to add missing clinical baseline variables for patients who were also enrolled in the MR CLEAN-NO IV trial. Although we regret that not all clinical variables are available for all patients included in our analysis, we feel that the data represented in Table 1 still provides an accurate overview of the included population as most patients had baseline data available.

We have updated Table 1 with the updated values. (Table 1)

1.7 Table 1: median follow up DWI time 23min? probably 23hrs...?

Indeed, this should be ‘hours’. We thank the reviewer for noticing and have adapted this accordingly. (Page 11-13, Table 1)

1.8 Line 307 duplication of "from"

Thank you. Corrected.

1.9 Suppl. figure S1 in my eyes does not add visual information, as I can visually barely detect sign. differences between the approaches.

We agree with the reviewer that it is hard to clearly see differences in S1 Fig and now see that this figure has limited added value. Hence, we have decided to remove S1 Fig from the Supporting Information and removed the references to S1 Fig from the manuscript. S2-S6 Figs have been renamed to S1-S5 Figs, accordingly.

Reviewer #2:

Overall interesting topic which needs further research as these thresholds are still to be validated and fu infarct volume is a good approximation.

2.1 One reference you might add to the discussion as it looks at a similar topic outlining the discrepancies between different software packages: https://pubmed.ncbi.nlm.nih.gov/33216157/

We thank the reviewer for this suggestion and have added the reference to both the Introduction and Discussion section. 

(Page 4, line 92-95): 

“The various commercially available CTP post-processing software packages use different approaches based on CBV or relative CBF (rCBF), and Tmax or rCBF parameters thresholds to estimate the respective ischemic core and penumbra, which complicates the generalizability of CTP results [8–11].”

(Page 19, line 517-519):

“Another study compared the core estimates from four software packages (i.e., RAPID, VEOcore, syngo.via, and Olea) and found volume differences up to 33 mL between the different software packages [11].”

---

## [Decision Letter · Decision Letter 1]

18 Jul 2022

Accuracy of CT perfusion ischemic core volume and location estimation: a comparison between four ischemic core estimation approaches using syngo.via

PONE-D-22-00274R1

Dear Dr. Hoving,

We’re pleased to inform you that your manuscript has been judged scientifically suitable for publication and will be formally accepted for publication once it meets all outstanding technical requirements.

Kind regards,

Miquel Vall-llosera Camps

Senior Editor

PLOS ONE

Reviewers' comments:

Reviewer's Responses to Questions

**Comments to the Author**

1. If the authors have adequately addressed your comments raised in a previous round of review and you feel that this manuscript is now acceptable for publication, you may indicate that here to bypass the “Comments to the Author” section, enter your conflict of interest statement in the “Confidential to Editor” section, and submit your "Accept" recommendation.

Reviewer #2: All comments have been addressed

2. Is the manuscript technically sound, and do the data support the conclusions?

Reviewer #2: Yes

3. Has the statistical analysis been performed appropriately and rigorously? 

Reviewer #2: Yes

4. Have the authors made all data underlying the findings in their manuscript fully available?

Reviewer #2: Yes

5. Is the manuscript presented in an intelligible fashion and written in standard English?

Reviewer #2: Yes

6. Review Comments to the Author

Reviewer #2: The Authors have sufficiently answered to all my comments and improved the manuscript very well.

7. PLOS authors have the option to publish the peer review history of their article (what does this mean?). If published, this will include your full peer review and any attached files.

Reviewer #2: No

---

## [Editor Report · Acceptance letter]

21 Jul 2022

PONE-D-22-00274R1 

Accuracy of CT perfusion ischemic core volume and location estimation: a comparison between four ischemic core estimation approaches using syngo.via 

Dear Dr. Hoving:

I'm pleased to inform you that your manuscript has been deemed suitable for publication in PLOS ONE. Congratulations! Your manuscript is now with our production department. 

Kind regards, 

on behalf of

Dr. Miquel Vall-llosera Camps 

Staff Editor

PLOS ONE